# Mitochondrial DNA: Hotspot for Potential Gene Modifiers Regulating Hypertrophic Cardiomyopathy

**DOI:** 10.3390/jcm9082349

**Published:** 2020-07-23

**Authors:** Parisa K. Kargaran, Jared M. Evans, Sara E. Bodbin, James G. W. Smith, Timothy J. Nelson, Chris Denning, Diogo Mosqueira

**Affiliations:** 1Department of Cardiovascular Medicine, Center for Regenerative Medicine, Mayo Clinic, Rochester, MN 55905, USA; Kargaran.Parisa@mayo.edu; 2Department of Health Science Research, Division of Biomedical Statistics and Informatics, Mayo Clinic, Rochester, MN 55905, USA; jaredmevans@outlook.com; 3Division of Cancer and Stem Cells, Biodiscovery Institute, University of Nottingham, Nottingham NG7 2RD, UK; paxsb7@exmail.nottingham.ac.uk; 4Faculty of Medicine and Health Sciences, Norwich Medical School, University of East Anglia, Norwich NR4 7UQ, UK; J.G.Smith@uea.ac.uk; 5Division of General Internal Medicine, Division of Pediatric Cardiology, Departments of Medicine, Molecular Pharmacology, and Experimental Therapeutics, Mayo Clinic Center for Regenerative Medicine, Rochester, MN 55905, USA; nelson.timothy@mayo.edu

**Keywords:** Hypertrophic cardiomyopathy, disease modeling, isogenic human pluripotent stem cell-derived cardiomyocytes, gene modifiers, mitochondrial DNA, haplogroups

## Abstract

Hypertrophic cardiomyopathy (HCM) is a prevalent and untreatable cardiovascular disease with a highly complex clinical and genetic causation. HCM patients bearing similar sarcomeric mutations display variable clinical outcomes, implying the involvement of gene modifiers that regulate disease progression. As individuals exhibiting mutations in mitochondrial DNA (mtDNA) present cardiac phenotypes, the mitochondrial genome is a promising candidate to harbor gene modifiers of HCM. Herein, we sequenced the mtDNA of isogenic pluripotent stem cell-cardiomyocyte models of HCM focusing on two sarcomeric mutations. This approach was extended to unrelated patient families totaling 52 cell lines. By correlating cellular and clinical phenotypes with mtDNA sequencing, potentially HCM-protective or -aggravator mtDNA variants were identified. These novel mutations were mostly located in the non-coding control region of the mtDNA and did not overlap with those of other mitochondrial diseases. Analysis of unrelated patients highlighted family-specific mtDNA variants, while others were common in particular population haplogroups. Further validation of mtDNA variants as gene modifiers is warranted but limited by the technically challenging methods of editing the mitochondrial genome. Future molecular characterization of these mtDNA variants in the context of HCM may identify novel treatments and facilitate genetic screening in cardiomyopathy patients towards more efficient treatment options.

## 1. Introduction

Hypertrophic cardiomyopathy (HCM) is a prevalent cardiovascular disease characterized by thickening of the left ventricular wall in the absence of abnormal loading conditions [1]. HCM progresses through a compensatory response relying on changes in gene expression, bioenergetics and cellular morphology of cardiomyocytes [2], in order to maintain the cardiac function required to meet the metabolic demands of the body [3]. HCM presents wide clinical variability, with patient outcomes ranging from asymptomatic to sudden cardiac death, when the sustained compensatory response leads to lethal energetic and functional imbalance [4]. This variation is likely due to the highly complex genetic causation characteristic of this condition, displaying variable penetrance and expressivity [5]. While approximately half of HCM patients display mutations in one or more of the >20 sarcomeric genes that regulate cardiomyocyte contraction, others do not exhibit any mutation in known heart-related genes [6]. Disease manifestation is mutation-specific, with patients with different mutations in the same gene showing contrasting clinical outcomes [7]. Even monozygotic twins bearing the same sarcomeric mutation displayed different clinical presentations [8], implying that factors beyond the single pathological change (e.g., genetic and epigenetic background and environmental modifiers) contribute to the magnitude of the disease effects [9]. This intricate clinical and genetic complexity of HCM has impaired the development of efficient therapeutics, with heart transplantation being the only long-term solution [10,11].

While numerous hallmarks of HCM have been identified [2], disrupted mitochondrial bioenergetics is commonly associated with heart failure, whereby the inefficient ATP usage in the heart leads to energy depletion [4]. Despite the fact that the majority of the proteins present in the mitochondria are encoded by nuclear genes, all the polypeptides, ribosomal RNAs and tRNAs encoded by mitochondrial DNA (mtDNA) constitute essential subunit complexes of the oxidative phosphorylation (OXPHOS) system [12]. The heart is a highly energetically-demanding organ that sources 90% of the ATP from mitochondrial OXPHOS [13]. Consistently, cardiomyocytes have higher mtDNA copy number per diploid nuclear genome, resulting in increased mitochondrial content [14]. Thus, mitochondrial diseases impact preferentially in the heart [15], mostly due to pathogenic mtDNA mutations [16] or reduction in mtDNA content [17]. Importantly, mitochondrial cardiomyopathies (MIC) constitute phenocopies of HCM, particularly in cases where the genes encoding mtRNAs are mutated [18]. Altogether, we hypothesize that mtDNA is a likely candidate for harboring gene modifiers of HCM, and it remains unexplored.

We previously developed and characterized parallel isogenic human pluripotent stem cell-derived cardiomyocyte (hPSC-CM) models of HCM, either displaying the p.R453C-β-myosin heavy chain (β-MHC) mutation [19], or the p.E99K-ACTC1 pathological change [20]. These two sets have revealed overall similarities and differences in HCM hallmarks and highlighted variations in functional phenotypes and gene expression profiles between the different cell lines carrying the same mutation. Remarkably, both sarcomeric-mutant models exhibited increased mitochondrial respiration rates relative to isogenic healthy controls, characteristic of the compensatory stage of HCM that precedes energy depletion [21]. Thus, we utilized these sets of hPSC-CMs to investigate the role of mtDNA to harbor potential gene modifiers that could underlie the different HCM phenotypes observed. The identification of novel HCM gene modifiers could unveil new molecular targets for future treatment and support genetic screening of patients, to better inform clinicians on the most efficient therapy.

## 2. Experimental Section

### 2.1. Cell Culture

hPSCs were cultured, characterized and differentiated into cardiomyocytes according to previously reported monolayer protocols [19,20,22] (all patient skin biopsies were donated via informed consent under the approval of Research Ethics Committee—number 09/H0408/74). hPSC-CMs at over ~90% purity were dissociated at Day 20 of culture by collagenase treatment, as previously described [23] and genomic DNA was extracted using DNeasy Blood and Tissue Kit (Qiagen, Hilden, Germany), following manufacturer′s instructions.

### 2.2. Ratiometric qPCR

Real-time qPCR reactions were performed via TaqMan^®^ Gene Expression assays (Applied Biosystems, Foster City, CA, USA), as previously described [24]. Briefly, 50 ng of genomic DNA was added to Taqman mastermix containing the probe of interest (NT-ND1: Hs02596873_s1, MT-ND2: Hs02596874_g1, ACTB: Hs03023880_g1). Data were analyzed using the 2^−ΔΔCT^ method to compare mtDNA content between healthy lines or the ΔΔC_T_ method between isogenic lines, where the average of healthy (WT) hPSC-CMs was used for relative quantification [25].

### 2.3. mtDNA Next Generation Sequencing (NGS) and Analysis

Whole mtDNA NGS was performed as previously described [26]. The reference sequence for mtDNA used was the revised Cambridge reference sequence (rCRS; GenBank Accession No. NC_012920.1). PCR integrity was confirmed with agarose gel imaging using two overlapping long-range PCR products (11,745 and 5277 bp) of the complete 16,569-bp mitochondrial genome, originating from a total DNA specimen. Sequencing was conducted using an Illumina HiSeq 4000 (Illumina, San Diego, CA, USA) with 51-bp paired-end reads. The Bioinformatics Core at Mayo Clinic analyzed the mtDNA NGS data using the mitosort pipeline. Briefly, the paired-end reads were aligned to the rCRS using BWA-MEM v0.7.10. A custom pysam method was used to call single nucleotide variants and small insertions/deletions with heteroplasmy greater than 1%. The Haplogrep2 method and PhyloTree 17 were used to assign haplogroups and identify the mtDNA private variants [27,28]. In silico prediction of the impact of mtDNA mutations in protein structure/function by Polyphen-2, CAROL and APOGEE was performed via the bioinformatics tool MitImpact3D (https://mitimpact.css-mendel.it/) [29]. The impact of mitochondrial tRNA mutations was evaluated by MITOTIP [30] (part of MITOMAP database: https://www.mitomap.org/MITOMAP).

### 2.4. Seahorse Analysis of Mitochondrial Respiration

The Seahorse XF96 extracellular flux analyzer (Agilent Technologies, Santa Clara, CA, USA) was used to evaluate cardiomyocyte metabolism, as previously described [24], with the values of oxygen consumption rate (OCR) being normalized to total cell number, quantified by Hoechst33342 (Sigma #BSS61) staining using fluorescence at 355 nm excitation and 460 nm emission in an automated imaging platform (CellaVista, Synentec, Elmshorn, Germany).

### 2.5. Evaluation of Production of Reactive Oxygen Species (ROS) in Mitochondria

Mitochondrial ROS production in live hPSC-CMs was investigated by flow cytometry after staining with 5 μM MitoSOX Red (#M36008 Life Technologies, Carlsbad, CA, USA) [31]. Live cells were incubated with the dye (made in Ca^2+^/Mg^2+^-rich HBSS (Gibco #14025, Thermo Fisher Scientific, Waltham, MA, USA)) for 30 min at 37 °C and 5% CO_2_, followed by a PBS wash (Gibco #14190-094, Thermo Fisher Scientific, Waltham, MA, USA) and centrifugation/ resuspension in PBS. Samples were then stored on ice for up to 30 min before being run on Astrios flow cytometer (Beckman Coulter, Brea, CA, USA).

### 2.6. Statistical Analysis

qPCR data are presented as box and whiskers plot representing 5–8 biological replicates performed in triplicate. Seahorse analysis and MitoSOX intensity is represented as mean ± SEM, from 5–7 biological replicates performed in triplicate. Statistical analysis was performed using GraphPad software (v8.2). Direct comparison between healthy cell lines was performed using unpaired Student’s *t*-test. HCM cell lines were compared to respective isogenic WT lines using unpaired one-way ANOVA test with Dunnett’s post hoc test for correction of multiple comparisons. Significance tests were based on *p*-values as follows: * *p* < 0.05; ** *p* < 0.01, *** *p* < 0.001, and n.s. (non-significant).

## 3. Results

### 3.1. Revisiting Isogenic hPSC-CM Models to Investigate Varying HCM Mitochondrial Phenotypes

To address the highly complex genetic causation associated with HCM, isogenic hPSC-CM lines were previously generated by CRISPR/Cas9 genome editing technology [32], which isolates the impact of the mutation of interest. This approach was used to introduce the p.R453C-β-MHC mutation [19] in three healthy (WT/WT) hPSC lines (AT1-hiPSC, REBL-PAT-hiPSC and HUES7-hESC), resulting in heterozygote (MUT/WT) and homozygote (MUT/MUT) variants, and a β-MHC-knockout (KO/KO) (Figure 1a) in a total of 10 cell lines. Previous characterization of the hPSC-CMs generated from these lines revealed HCM hallmarks of disease (e.g., hypertrophy, hypo-contractility and bioenergetics), but to a different degree between sets, with the AT1 lines showing the more severe phenotypes relative to REBL-PAT and HUES7 counterparts [19].

In addition, isogenic hiPSC sets bearing the heterozygous p.E99K-ACTC1 mutation were generated by correcting patient lines (E99K1 and -2) or introducing the mutation in healthy relatives (NC-Edit-E99K) [20]. Similar to the β-MHC-mutant model, the magnitude of phenotypes investigated (e.g., arrhythmogenesis, hyper-contractility and energy depletion) varied between the cell line of origin (with E99K1 line showing more severe phenotypes than E99K2 and NC-Edit-E99K). In light of the strong contribution of energy depletion to HCM progression in both models, the mtDNA sequence and content were investigated and data were related to the differential functional phenotypes previously observed between separate isogenic sets bearing the same mutation.

A ratiometric qPCR approach was undertaken to assess mitochondrial content in isogenic p.R453C-β-MHC hPSC-CMs, by analyzing mitochondrial: nuclear DNA ratios [24]. MtDNA content varied between healthy hPSC-CM lines (Figure 1b), following the same trend as the HCM phenotypic severity: AT1-cardiomyocytes showed markedly lower mtDNA content relative to REBL-PAT and HUES7 counterparts. Introduction of the p.R453C-β-MHC mutation in AT1 and REBL-PAT hPSC-CMs did not alter mtDNA content relative to their healthy isogenic controls, except for the most extreme genotypes (AT1-β-MHC-KO and partially in REBL-PAT homozygous, Figure 1c,d).

### 3.2. Comparison of mtDNA Variants between Phenotyped Isogenic Lines Reveals Potential HCM Gene Modifiers

As hPSC-CM lines bearing the same mutation showed variations in magnitude of HCM phenotypes, we hypothesized that these differences could be due to mtDNA mutations tracing back to the cell source of origin used for disease modeling. Thus, whole mtDNA next-generation sequencing (NGS) was performed for all the cell lines investigated (from fibroblasts through hPSCs to differentiated hPSC-CMs). The identification of mtDNA variants through nuclear reprograming and cardiomyocyte differentiation enables exclusion of mutations that arise as a result of in vitro culture [26], i.e., that do not play a role in HCM progression.

Analysis of mtDNA NGS data using HaploGrep software revealed clear differences between the three starting hPSC lines used for p.R453C-βMHC HCM modeling, consistent between all the lineages (Figure 2). These encompassed different types of mutations (deletions (DELs), insertions (INSs) and single nucleotide variants (SNVs)) and heteroplasmy, with some only being introduced or lost upon in vitro culture (and thus deemed artefactual for HCM). To predict the impact of these mutations in protein structure/function, we utilized PolyPhen-2, CAROL and APOGEE, as these were among the top in silico prediction performers [29,33]. In addition, the impact of Mt-tRNA variants was predicted by MITOTIP [30]. However, all of the potentially HCM-relevant mutations were located in the non-coding control region of the mtDNA genome (Table 1), preventing in silico prediction.

To understand the contribution of the identified mutations to HCM progression, the heteroplasmy levels between hPSC-CMs were related to the known phenotype severities (AT1 more severe than REBL-PAT and HUES7 [19]). Interestingly, some mutations were only present in the AT1 samples (m.152T > C, m.514_515delCA) deeming them as potential aggravators. Contrarily, others were only identified in REBL-PAT and HUES7 lines (m.309_310insCCT, m.309_310insCCCT and m.16319G > A), rendering them potential protective gene modifiers. Lastly, some variants were present in all lines, with varying heteroplasmy levels following a clear trend (e.g., m.310T > C, lower in AT1 than REBL-PAT and HUES7—potentially protective; m.310_311insC, higher in AT1 than REBL-PAT and HUES7—likely aggravator).

The same strategy was applied to the three isogenic pairs (E99K1, E99K2 and NC-Edit-E99K) bearing the heterozygous p.E99K-ACTC1 mutation. Analysis of NGS of mtDNA highlighted different variants between lines (Figure 3), with the majority being located in the non-coding control region (Table 2). This approach identified a variant exclusively in the E99K1 line (m.11176G > A), with another being shared with the AT1 line bearing the p.R453C-βMHC sarcomeric mutation (m.152T > C), making both strong candidates as potential HCM aggravators. Remarkably, some heteroplasmic mutations were common between the most severe p.E99K-ACTC1 (E99K1 line) and the milder p.R453C-β-HMC models (REBL-PAT and HUES7 lines), suggesting opposite potential effect in HCM progression: m.309_310insCCT, m.309_310insCCCT, m.310T > C and m.310_311insC. Finally, some variants were only detected in the NC-Edit-E99K and E99K2 lines (m.8952T > C; m.12715A > G), where HCM phenotypes were less acute or non-existent [20], indicating a potentially cardioprotective modifier function. The identified correlations between mtDNA variants and phenotypic severity across multiple isogenic hPSC-CM lines bearing different sarcomeric mutations strongly support the role of the non-coding control region to harbor gene modifiers of HCM.

To test whether these mtDNA variants had a non-specific mito-modulating effect, mitochondrial respiration was evaluated in the healthy hPSC-CM lines under study using the Seahorse platform (Figure 4a–d). Interestingly, the cell lines showing less pronounced phenotypes upon the introduction of the sarcomeric mutations (i.e., REBL-PAT and NC) showed higher baseline mitochondrial respiration activity relative to more affected hiPSC-CM lines (AT1 and E99K1), in line with the higher mtDNA content displayed (Figure 1b). This highlights the role of mtDNA variants to HCM progression: while the identified mutations were not the cause of the cellular phenotypes previously observed (as they are shared between healthy and diseased hPSC-CMs), they contribute to varying baseline levels of OXPHOS in healthy mitochondria, suggesting their involvement as gene modifiers. The isogenic approach used herein enables the investigation of mtDNA mutations shared by unrelated cell lines (in addition to variants in the nuclear genome, which were not addressed in this study).

In addition, we investigated whether the identified mtDNA variants were associated with global mito-dysfunction, by performing live cell flow cytometry of 5 μM MitoSOX-labeled hPSC-CMs, which indicates the presence of reactive oxygen species (ROS) in the mitochondria [31]. Elevated mitochondrial ROS levels cause damage to biological macromolecules and mtDNA mutations, as the mitochondrial genome is in close proximity of the inner membrane where ROS are produced [34]. However, analysis of the mean intensity of MitoSOX showed similar levels between healthy AT1 and REBL-PAT hPSC-CMs (Figure 4e,f), indicating that global mito-dysfunction was neither associated with the differences in magnitude in the cellular phenotypes between lines nor with the existence of different mtDNA variants between them.

### 3.3. MtDNA Sequencing Shows Common Variants in Unrelated HCM Patients

To improve our understanding of variability of HCM outcomes between patients, we extended the investigation of the p.E99K-ACTC1 lines to three unrelated families where that sarcomeric mutation was predominant. Skin punch biopsies were obtained from 13 individuals (Figure 5a–c), comprising 3 healthy (Samples NC, 10 and 14, in green) and 10 HCM patients, three of which showed more severe in vitro phenotypes (sample E99K1) or NYHA classification II (Samples 9 and 13, in red), while the remainder were classed as NYHA-I (in blue). NGS of mtDNA derived from isolated fibroblasts from these patients reflected expected maternal inheritance patterns (e.g., Sample 7 descending from Sample 6 and Samples 16 and 18 inheriting from Sample 17, Figure 5d), and revealed different mutations between individuals. In-depth analysis of these mtDNA variants (Appendix A) showed that, while some mutations were present in all three families at different heteroplasmy levels (m.310T > C, m.310_311insC and m.5597A > C), others were common between two (e.g., m.310_311insC and m.513_514insCACA) or even restricted to some members of only one family (e.g., m.9025G > A). The fact that specific mtDNA mutations are shared between unrelated families with similar clinical severity further supports the involvement of mtDNA in HCM progression.

### 3.4. The Identified mtDNA Mutations Are Novel for HCM, But May Be Common in Some Haplogroups

The contribution of mtDNA variants to cardiomyopathy progression remains vastly unexplored, with most of the information deriving from mitochondrial neuromuscular diseases [17]. To further ascertain whether the mtDNA mutations identified in the hPSC-CM lines and patient families contribute to HCM, we compared them with the few existing reports where the mtDNA of patients presenting cardiac phenotypes was sequenced [18,26,35,36]. In addition, we cross-referenced the most comprehensive bioinformatics database of mtDNA variants associated with human disease (https://www.mitomap.org/MITOMAP). Remarkably, there was no overlap between the mtDNA mutations identified herein and those detected in patients with abnormalities in cardiac function or exhibiting any other mitochondrial diseases.

As none of the identified mtDNA mutations were previously detected in any disease, we further utilized MITOMAP [37] to investigate the prevalence of these polymorphisms in the human population and/or within haplogroups. This database entails over 50,000 full-length mtDNA sequences and over 73,000 control region mtDNA sequences. We focused on the mtDNA mutations detected across the hPSC-CM lines that suggested modifier effects on HCM (Table 3). The detected variants do not belong to each sample haplogroup and were classified by HaploGrep depending on whether or not they are found in other haplogroups as “local” or “global” private mutations, respectively [26].

These ten mutations showed variable frequency in the mtDNA sequences deposited in the MITOMAP database, ranging from rare (m.309_310insCCCT; m.310_311_insC; m.8952T > C; m.12715A > G) to more prevalent (m.152T > C; m.309_310insCT; m.310T > C) incidence. However, half of them are enriched (over 50% frequency) in specific haplogroups (Appendix A). Importantly, clinical studies have indicated that the haplogroup H is found in higher frequency in HCM probands, indicating a susceptibility factor, whereas haplogroups J, K and UK are uncommon in patients, representing protective factors [38]. Remarkably, all the variants identified in hPSC-CMs as potential HCM modifiers were enriched in haplogroup H, with variants m.152T > C and m.310T > C also being common in haplogroups J and K. This overlap between the cellular model and patient cohort data further supports our hypothesis that the identified mtDNA variants may contribute as risk modifiers for HCM. This suggests that HCM patients belonging to these haplogroups are likely to show different clinical outcomes, depending on whether the mtDNA mutation they present is a potential aggravator or protective to the disease progression.

## 4. Discussion

To achieve a deeper understanding of pathological mechanisms of HCM, several models of disease have been developed, ranging from preparations of scarcely available human heart tissue biopsies to whole animals [39]. Recent technological advances have resulted in the application of hPSC-CMs for disease modeling, advantageous relative to previous models due to the ability to provide an unlimited source of cardiomyocytes and recapture patient genetic makeup [40]. Genome editing technologies such as CRISPR/Cas9 have facilitated the generation of high-fidelity HCM models, surpassing the inherent genetic and epigenetic baseline variation between unrelated hPSC-CM lines [41]. Isogenic sets of hPSC-CM models have therefore addressed the complex genetic etiology characteristic of HCM and corroborated clinical observations, highlighting that disease progression is mutation-specific [21] and dependent on factors beyond the primary mutation [20,42].

Early approaches aiming to establish genotype-phenotype relationships in HCM relied heavily on high numbers of patients in order to associate genetic variants with disease severity, due to the existence of confounding factors such as age and donor variability [43]. Remarkably, the use of multiple isogenic cell lines sharing the same sarcomeric mutation in different genetic backgrounds has complemented patient cohort studies by facilitating the determination of the penetrance and expressivity associated with that variant. Importantly, the controlled genetic background of this approach overcomes the need for high numbers of patient cohorts and enables the identification of novel gene modifiers of HCM that may constitute new therapeutic targets. We have contributed to this goal by generating and characterizing two models of HCM bearing different sarcomeric mutations (p.R453C-βMHC or p.E99K-ACTC1) in six unrelated healthy hPSC cell lines [19,20]. These two models exhibited the main functional hallmarks of HCM (hypertrophy, energy depletion, arrhythmias and changes in contractility) and showed variation in the magnitude of in vitro cellular phenotypes observed between isogenic sets sharing the same mutation. Due to the fact that energetic imbalances in the heart trigger heart failure in HCM patients [4] and that genetic abnormalities in mtDNA can recapitulate cardiomyopathy phenotypes [16], we investigated mtDNA as a likely candidate for harboring gene modifiers of HCM.

While sequencing mtDNA of a large cohort of patients would have been advantageous to provide a higher degree of patient-wide validation, the alternative use of isogenic sets of hPSC-CM lines enables a more refined investigation of the contribution of mtDNA mutations to HCM, a disease characterized by highly complex mutation-specific effects [21]. Additionally, the customary investigation of patient peripheral blood as an alternative to scarcely available cardiac tissue biopsies is not optimal due to tissue-specific mtDNA sequence heteroplasmy [44] (which is recapitulated in hPSC-CMs). Notwithstanding, the use of cell lines instead of patient DNA for mitochondrial genetics studies also presents some drawbacks such as unintended mtDNA mutagenesis and changes in heteroplasmy that may arise during reprogramming, long-term culture, and cardiac differentiation [26]. Nevertheless, these technical artefacts were minimized by: (i) applying the same culture conditions across all the cell lines under study; (ii) directly comparing isogenic cell lines where the genetic background is preserved (impossible in patient samples); and mostly by (iii) sequencing all the lineages from patient-derived fibroblasts, through hPSCs and hPSC-CMs to identify HCM-unspecific mtDNA mutations. Additionally, we extended our findings from hPSC-CMs to three unrelated patient families exhibiting the p.E99K-ACTC1 mutation, providing further evidence that NGS of cell lines may complement that of patient DNA.

NGS analysis of mtDNA derived from isogenic sets of healthy and HCM hPSC-CMs revealed 11 variants in the p.R453C-βMHC model and 10 in the p.E99K-ACTC1 set. However, in silico prediction combined with sequencing all the lineages from fibroblasts to hPSCs and hPSC-CMs resulted in the exclusion of four in the former model and two in the latter. By correlating previously reported functional disease phenotypes [19,20] with variant heteroplasmy analyzed herein, we identified potential HCM aggravators and protective mtDNA mutations in both models. Typically, cells can withstand a high proportion of mutant:WT mtDNA ratios, but when that proportion exceeds the disease thresholding effect (specific for each variant), OXPHOS defects occur [45].

Moreover, some mtDNA mutations were complete, totally abolishing WT copies. While some of such variants were specific to each model (such as m.12715A > G as potentially cardioprotective), one was shared between them (m.152T > C) as a highly likely disease aggravator. Conversely, four variants showed opposing potential effects in the two models, reinforcing the notion that HCM exhibits complex genetic causation. We speculate that these variants reflect the contrasting changes in phenotype induced by the two primary sarcomeric mutations, e.g., while the p.R453C-βMHC change caused hypo-contractility, the p.E99K-ACTC1 mutation led to hyper-contractility [21].

Importantly, some of the variants identified in hPSC-CMs were enriched in specific haplogroups, complementing patient cohort studies that highlighted the haplogroup H as a susceptibility factor due to its higher prevalence in HCM patients [38]. As the identification of the specific mtDNA variants underlying these differences is lacking [46], further validation of their role in HCM in hPSC-CMs may contribute to refining this association.

To extend the drawn correlations to a wider patient population, three unrelated families where p.E99K-ACTC1 (and consequently HCM) were common were investigated, unveiling a number of shared or family-exclusive mtDNA mutations that may underlie variation in clinical outcomes. However, when compared to the bioinformatics databases focused on human diseases caused by mtDNA mutations, none of the identified variants were overlapping. Moreover, MIC patients presenting cardiomyopathy phenotypes have not showed mutations in the non-coding control region of mtDNA [18]. This implies that whilst mtDNA mutations are central to MIC and constitute the main cause of disease, they are not sufficient to induce HCM, unlike sarcomeric mutations. Nonetheless, they can exacerbate or improve clinical outcomes, as was illustrated in more severe hypertrophy phenotypes (~72% higher intraventricular septum thickness) in patients showing both sarcomeric (p.R249Q-βMHC) and mitochondrial (m.4300A > G) mutations relative to those bearing only one of them [47]. The same applies to changes in mitochondrial content: while MIC patients showed striking phenotypes in terms of maladaptive mitochondrial proliferation (e.g., ~3-fold increase in mtDNA content per cell relative to healthy samples [48]), no conclusive evidence has shown variations in mtDNA content in explanted failing hearts when compared to healthy counterparts [49], in line with our data from hPSC-CMs.

To fully demonstrate their role in disease progression, gain- or loss-of-function studies should be performed, whereby the mtDNA mutations are corrected or introduced in hPSC-CMs. This was not done in the scope of this study because precise genomic correction of mtDNA mutations is technically very challenging and has not yet been achieved robustly and efficiently [50,51]. CRISPR/Cas9 approaches rely on homologous recombination (HR) of double stranded breaks (DSBs) to introduce SNVs of interest [52]. However, unlike in the nuclear DNA, DSBs in mammalian mtDNA cannot be efficiently repaired by HR [53], precluding the use of CRISPR/Cas9 [54]. mtDNA molecules bearing DSBs are rapidly degraded and the remaining intact molecules replicate to restore a steady mtDNA copy number, shifting the heteroplasmic ratio [55]. This strategy has been explored by engineering restriction endonucleases directed to the mitochondria [56,57] to degrade mutant mtDNA. While this approach may be partially feasible in the case of heteroplasmic mutations, it is ineffective for those where the total mtDNA is mutated (such as m.152T > C and m.16319G > A identified in this study). Alternatively, another possibility is the delivery of healthy plasmid DNA to the mitochondria. However, the efficient transport of nucleic acids into this organelle remains elusive, even when conjugated with mitochondriotropic delivery systems [58]. Clearly, a technological breakthrough is required for efficient and precise genomic correction of mtDNA mutations in MIC and HCM patients.

## 5. Conclusions

Taken together, by correlating mtDNA variant heteroplasmy with observed in vitro phenotypes and clinical information, this study identified a number of novel gene modifiers that may potentially contribute to the clinical outcome of HCM. In addition, specific patient haplogroups bearing the identified mtDNA variants may be at a higher or lower risk of progressing HCM to heart failure. This adds another layer of complexity to efficient treatment decision: not only should clinicians consider sarcomeric mutation-specific effects, but also the existence of gene modifiers that could exacerbate or ameliorate clinical outcomes. This raises the possibility of performing genetic screening for mtDNA mutations, which is also useful to distinguish between HCM and MIC patients (particularly in cases with unexplained hypertrophy without fulfilling standard criteria for HCM, such as symmetrical hypertrophy with absence of left ventricular outflow tract obstruction) [1]. Overall, this report illustrates the involvement of hitherto unexplored modifiers of HCM, paving the way for future functional studies addressing the high complexity of this cardiovascular disease.

## Figures and Tables

**Figure 1 jcm-09-02349-f001:**
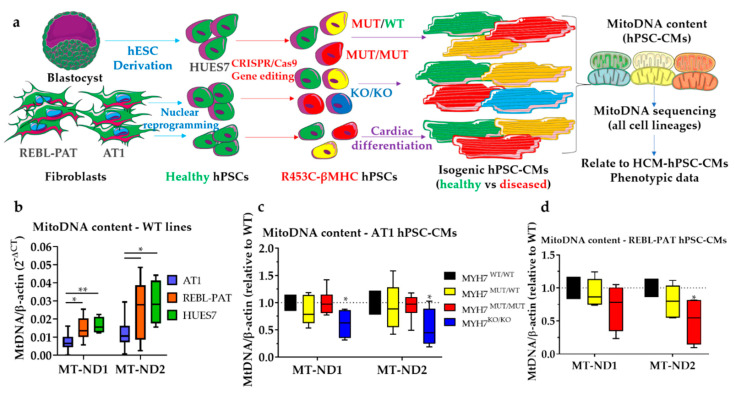
Harnessing isogenic hPSC-CM models to investigate varying HCM mitochondrial phenotypes. (**a**) Study workflow: hPSC were generated from three different healthy cell sources, followed by the introduction of the p.R453C-βMHC mutation by CRISPR/Cas9. Isogenic healthy and diseased (heterozygote or homozygote) mutant hPSCs were differentiated into cardiomyocytes and mtDNA content and sequence was evaluated. (**b**) Ratiometric qPCR analysis of mitochondrial: nuclear DNA highlighted variations in mitochondrial content across healthy (WT) lines, with AT1 lines showing 50–60% lower mtDNA content relative to HUES7 and REBL-PAT hPSC-CMs. Isogenic (**c**) AT1 and (**d**) REBL-PAT hPSC-CMs did not display striking changes in mtDNA content, with the exception of the AT1 MYH7-knockout line. Data are presented as box and whiskers plots, n = 5–8 biological replicates, * *p* < 0.05, ** *p* < 0.01 (Student’s t-test in (**b**) and one-way ANOVA + Dunnett’s correction relative to WT in (**c**,**d**)).

**Figure 2 jcm-09-02349-f002:**
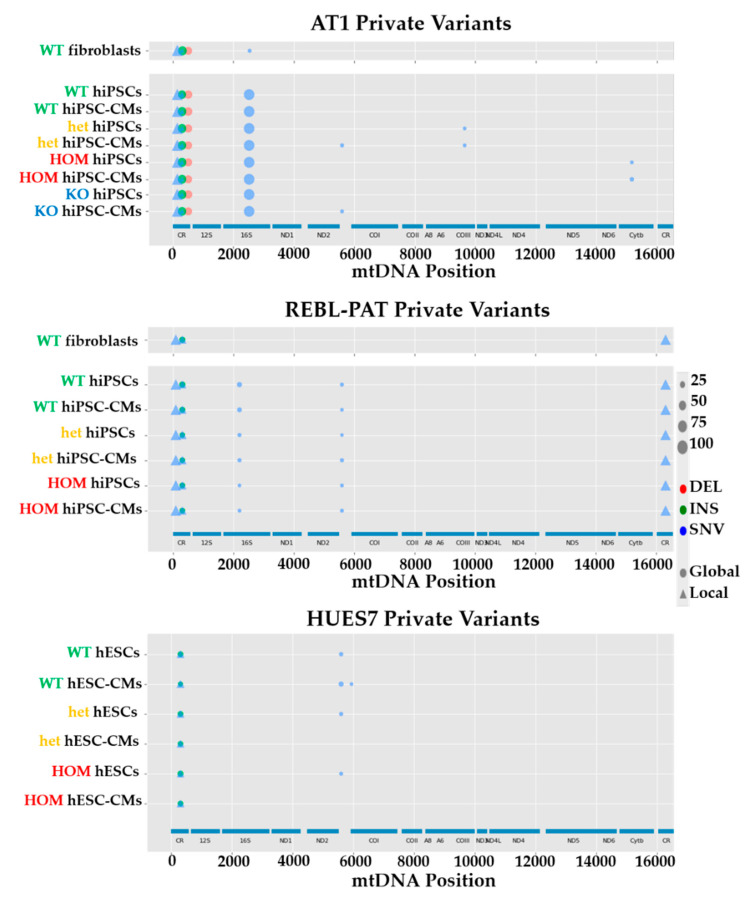
mtDNA NGS of three separate isogenic sets bearing the p.R453C-βMHC mutation. Analysis of mtDNA sequence of isogenic sets of fibroblasts, hPSCs and hPSC-CMs of AT1 (top), REBL-PAT (middle) and HUES7 (bottom) identified variants specific to the starting cell sources. Legend map on the right side indicates type of mtDNA mutation (DEL, deletion; INS, insertion; SNV, single nucleotide variant) and percent heteroplasmy is correlated with the size of the symbol (WT, wild-type; het, heterozygous; HOM, homozygous; KO, knock-out).

**Figure 3 jcm-09-02349-f003:**
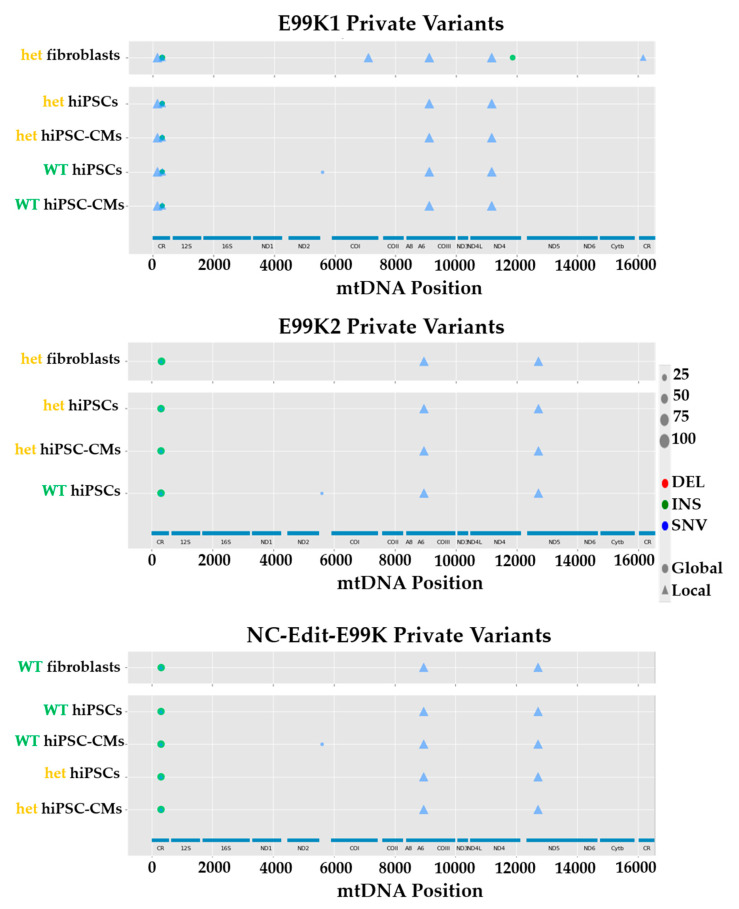
mtDNA NGS of three separate isogenic sets bearing the p.E99K-ACTC1 mutation. Analysis of mtDNA sequence of isogenic sets of fibroblasts, hPSCs and hPSC-CMs of E99K1 (top), E99K2 (middle) and NC-Edit-E99K (bottom) identified variants specific to the starting cell sources. Legend map on the right side indicates type of mtDNA mutation (DEL, deletion; INS, insertion; SNV, single nucleotide variant) and percent heteroplasmy is correlated with the size of the symbol. (WT, wild-type; het, heterozygous).

**Figure 4 jcm-09-02349-f004:**
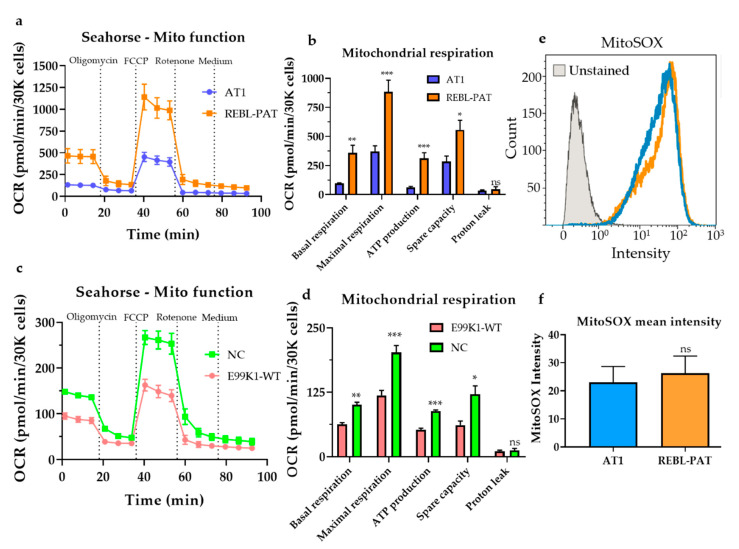
Mito-modulating effects of mtDNA variants identified in hPSC-CMs. (**a**) Mitochondrial respiration was evaluated by Seahorse analysis of healthy hPSC-CM lines used to model the p.R453C-βMHC sarcomeric mutation, showing (**b**) higher oxidative phosphorylation activity in REBL-PAT cardiomyocytes relative to the AT1 line, presenting different mtDNA variants. (**c**,**d**) The healthy (gene-corrected) E99K1-WT hPSC-CM line showed lower mitochondrial respiration activity relative to NC hPSC-CMs displaying diverse mtDNA variants. (**e**,**f**) Evaluation of mitochondrial ROS production by flow cytometry of MitoSOX-labelled hPSC-CMs showed no difference between healthy AT1 vs REBL-PAT hPSC-CMs, indicating that the mtDNA variants identified in these lines do not contribute to mito-dysfunction. Data are presented as mean ± SEM, N = 5-7 biological replicates, * *p* < 0.05, ** *p* < 0.01, *** *p* < 0.001, n.s., non-significant (Student’s *t*-test); OCR, oxygen consumption rate.

**Figure 5 jcm-09-02349-f005:**
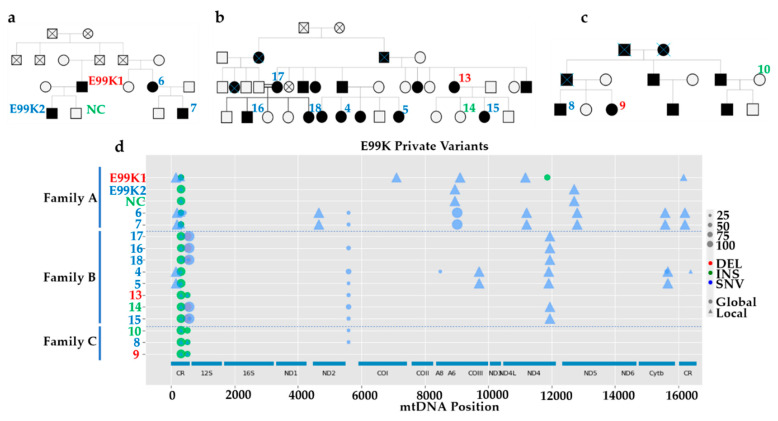
Analysis of mtDNA variants in unrelated HCM patients. (**a**–**c**) Pedigree chart of three unrelated families where the p.E99K-ACTC1 mutation is common (red numbers indicate more severe phenotypes and green numbers milder ones). (**d**) NGS sequencing analysis of mtDNA of fibroblasts derived from patient skin biopsies identified variants shared between patients or specific to individual families. Legend map on the right side indicates type of mtDNA mutation (DEL, deletion; INS, insertion; SNV, single nucleotide variant) and percent heteroplasmy is correlated with the size of the symbol.

**Table 1 jcm-09-02349-t001:** mtDNA variants identified by NGS in p.R453C-β-MHC isogenic lines used for HCM modeling (WT, healthy; het, heterozygous; HOM, homozygous; KO, knockout). In silico pathogenicity prediction algorithms used: ^§^ MitoTIP; * Polyphen-2; ^†^ CAROL; ^‡^ APOGEE. Phenotypically severe hPSC-CM line is highlighted in red and less severe ones in green.

Isogenic hPSC-CM Set(s)	Locus	mtDNA Mutation	Percent Heteroplasmy (WT-Het-HOM-KO)	In silico Prediction (Score)	Potential HCM Effect
AT1	MT-HV2, MT-OHR	m.152T > C	99.9 (all lines)	N/A (non-coding)	Aggravator (present only in AT1s)
REBL-PAT + HUES7	MT-HV2, MT-OHR, MT-CSB2	m.309_310insCCT	REBL-PAT: 28.9–26.4–27.2 HUES7: 17.4–13.8–15.6	N/A (non-coding)	Protective (absent in AT1)
HUES7	MT-HV2, MT-OHR, MT-CSB2	m.309_310insCCCT	11.7–11.6–13.3	N/A (non-coding)	Protective (absent in AT1)
AT1 + REBL-PAT + HUES7	MT-HV2, MT-OHR, MT-CSB2	m.310T > C	AT1: 15.0–17.1–14.5–17.0; REBL-PAT: 61.2–63.2–61.9; HUES7: 61.6–56.4–57.6	N/A (non-coding)	Protective (less prevalent in AT1)
AT1 + REBL-PAT + HUES7	MT-HV2, MT-OHR, MT-CSB2	m.310_311insC	AT1: 67.8–64.2–65.6–63.9 REBL-PAT: 30.1–26.7–27.2 HUES7: 26.1–32.0–33.3	N/A (non-coding)	Aggravator (more prevalent in AT1s)
AT1	MT-HV3	m.514_515delCA	60.1–53.5–53.6–53.7;	N/A (non-coding)	Aggravator only present in AT1s)
REBL-PAT	MT-RNR2	m.2203G > A	14.5%	N/A (non-coding)	None: acquired during reprogramming
AT1	MT-RNR2	m.2525C > T	Fibroblasts: 12.1% iPSCs + iPSC-CMs: 99.9%	N/A (non-coding)	None: increased upon nuclear reprogramming
AT1 + REBL-PAT + HUES7	MT-TA	m.5597A > C	AT1: 5.4–12.4–6.5–11.4; REBL-PAT: 11.1–14.3–13.2; HUES7: 24.8–7.8–8.8	Possibly pathogenic (73.80%) ^§^	None: acquired during reprogramming/culture
HUES7	MT-CO1	m.5938A > C	12.2–1.2–1.1	Probably damaging (1.000) * Deleterious (1) ^†^ Neutral (0.45) ^‡^	None: acquired during culture
REBL-PAT	MT-HV1	m.16319G > A	99.7–99.7–99.6	N/A (non-coding)	Protective (absent in AT1)

**Table 2 jcm-09-02349-t002:** mtDNA variants identified by NGS in p.E99K-ACTC1 isogenic lines used for HCM modeling (WT, healthy; het, heterozygous). In silico pathogenicity prediction algorithms used: ^§^ MitoTIP; * Polyphen-2; ^†^ CAROL; ^‡^ APOGEE. Phenotypically severe hPSC-CM line is highlighted in red and less severe ones in green.

Isogenic hPSC-CM Set(s)	Locus	mtDNA Mutation	Percent Heteroplasmy (WT-Het)	In silico Prediction (Score)	Potential HCM Effect
E99K1	MT-HV2, MT-OHR	m.152T > C	99.3–99.9	N/A (non-coding)	Aggravator (present only in E99K1)
E99K1	MT-HV2, MT-OHR, MT-CSB2	m.309_310insCT	15.3–25.3	N/A (non-coding)	Aggravator (present only in E99K1)
E99K1	MT-HV2, MT-OHR, MT-CSB2	m.309_310insCCT	18.8–9.3	N/A (non-coding)	Aggravator (present only in E99K1)
E99K1 + NC-EDIT-E99K + E99K2	MT-HV2, MT-OHR, MT-CSB2	m.310T > C	E99K1: 60.6–59.9 NC-EDIT-E99K: 17.0–16.3 E99K2: 17.5–16.4	N/A (non-coding)	Aggravator (more prevalent in E99K1)
E99K1 + NC-EDIT-E99K + E99K2	MT-HV2, MT-OHR, MT-CSB2	m.310_311insC	E99K1: 26.6–27.2 NC-EDIT-E99K: 61.6–62.7 E99K2: 63.6–61.7	N/A (non-coding)	Protective (less prevalent in E99K1)
E99K1 + NC-EDIT-E99K + E99K2	MT-TA	m.5597A > C	E99K1: 9.6–9.6 NC-EDIT-E99K: 7.5–6.9 E99K2: 10.2–7.1	Possibly pathogenic (73.80%) ^§^	None (present in all lines)
NC-EDIT-E99K + E99K2	MT-ATP6	m.8952T > C	NC-EDIT-E99K: 99.9–99.9 E99K2:100–99.9	N/A (redundant)	Protective (absent in E99K1);
E99K1	MT-ATP6	m.9116T > C	100–99.5	Benign (0.000) * Neutral (0.8) ^†^ Neutral (0.42) ^‡^	None (benign mutation)
E99K1	MT-ND4	m.11176G > A	99.9–99.2	N/A (redundant)	Aggravator (only present in E99K1)
NC-EDIT-E99K + E99K2	MT-ND5	m.12715A > G	NC-EDIT-E99K: 99.8–99.8 E99K2: 99.9–99.7	Benign (0.1) * Neutral (0.47) ^†^ Neutral (0.28) ^‡^	Protective (absent in E99K1)

**Table 3 jcm-09-02349-t003:** Prevalence of mtDNA mutations identified in hPSC-CMs using MITOMAP database. Green, potential HCM protective variants; red, potential HCM aggravator variants.

mtDNA Mutation	Variant Type	Percent Variant Frequency (Full Length Sequences)	Percent Variant Frequency (Control Region Sequences)	Found at Haplogroups at 50% or Higher
m.152T > C	Local	26.5	17.6	Yes
m.309_310insCT	Global	26.4	16.4	Yes
m.309_310insCCT	Global	7.1	4.9	Yes
m.309_310insCCCT	Global	0.1	0.2	No
m.310T > C	Local	40.7	27.8	Yes
m.310_311_insC	Global	0.0	0.0	No
m.514_515delCA	Global	24.2	1.7	No
m.8952T > C	Local	0.0	N/A	No
m.12715A > G	Local	0.1	N/A	No
m.16319G > A	Local	5.9	8.0	Yes

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
