# Peer review of "Mitochondrial DNA: Hotspot for Potential Gene Modifiers Regulating Hypertrophic Cardiomyopathy"

_jcm, 2020, doi:10.3390/jcm9082349_

Round 1

Reviewer 1 Report

The paper by Kargaran et al sequenced mtDNA from patient cell lines and noted mtDNA variants that appear to modify hypertrophic cardiomyopathy disease severity.  Most of the variants identified seem to be in the regulatory D-loop and appear to be very rare

Specific comments

  1. Use of cell lines rather than patient DNA adds a layer of potential error or artifact (this should be discussed)
  2. mtDNA sequencing of host DNA in a large cohort of patients versus controls would have been preferred
  3. Mitochondrial function in the cells (e.g. SeaHorse) could be a way to assess if D-loop variants have a non-specific mito-modulating effect.
  4. Similarly, evidence for mito-dysfunction (e.g. HgbA1c levels in the patients) would hint towards global effects on mitochondria
  5. What is the frequency of cardiac disease in the haplogroups affected by the variants?
  6. Conclusion recommending mtDNA sequencing in the clinical setting is premature

Reviewer 2 Report

The authors sequenced the mtDNA of isogenic pluripotent stem cell

cardiomyocyte models of HCM focusing on two sarcomeric mutations.

By correlating cellular and clinical phenotypes with mtDNA sequencing, potentially HCM-protective or aggravator mtDNA variants were identified.  These novel mutations were mostly located in the non-coding control region of the mtDNA and did not overlap with those of other mitochondrial diseases. Analysis of unrelated patients highlighted family-specific mtDNA variants, while others were common in particular population haplogroups. Further validation of mtDNA variants as gene modifiers is warranted but limited by the technically challenging methods of editing the mitochondrial genome. The authors propose mtDNA sequencing of HCM patients as a criterion to better inform clinicians on the disease

Although further validation of mtDNA variants as gene modifiers is warranted, the findings are clinically interesting and an important contribution to the literature. To further strengthen the manuscript, there is a concern with this paper that should be addressed before considering publication.

Results

Page 4, lines 163-166, Table 1

In order to predict the impact of these mutations in protein structure/function, PolyPhen-2 was utilized.

Are there any other methods that authors used to predict the impact of these mutations in protein structure/function?  Several methods should be used to predict impact of these mutations in protein structure/function.
